# Theoretical Approach for the Luminescent Properties of Ir(III) Complexes to Produce Red–Green–Blue LEC Devices

**DOI:** 10.3390/molecules27092623

**Published:** 2022-04-19

**Authors:** Mireya Santander-Nelli, Bastián Boza, Felipe Salas, David Zambrano, Luis Rosales, Paulina Dreyse

**Affiliations:** 1Advanced Integrated Technologies (AINTECH), Chorrillo Uno, Parcela 21, Lampa, Santiago 9390015, Chile; 2Centro Integrativo de Biología y Química Aplicada (CIBQA), Universidad Bernardo O’Higgins, General Gana 1702, Santiago 8370854, Chile; 3Departamento de Química, Universidad Técnica Federico Santa María, Avda. España 1680, Casilla, Valparaíso 2390123, Chile; bastian.boza.14@sansano.usm.cl (B.B.); felipe.salasc@usm.cl (F.S.); 4Departamento de Física, Universidad Técnica Federico Santa María, Avda. España 1680, Casilla, Valparaíso 2390123, Chile; david.zambrano@usm.cl (D.Z.); luis.rosalesa@usm.cl (L.R.)

**Keywords:** Ir-iTMC, LEC devices, TD-DFT, phosphorescence, RGB

## Abstract

With an appropriate mixture of cyclometalating and ancillary ligands, based on simple structures (commercial or easily synthesized), it has been possible to design a family of eight new Ir(III) complexes (**1A**, **1B**, **2B**, **2C**, **3B**, **3C, 3D** and **3E**) useful as luminescent materials in LEC devices. These complexes involved the use of phenylpyridines or fluorophenylpyridines as cyclometalating ligands and bipyridine or phenanthroline-type structures as ancillary ligands. The emitting properties have been evaluated from a theoretical approach through Density Functional Theory and Time-Dependent Density Functional Theory calculations, determining geometric parameters, frontier orbital energies, absorption and emission energies, injection and transport parameters of holes and electrons, and parameters associated with the radiative and non-radiative decays. With these complexes it was possible to obtain a wide range of emission colours, from deep red to blue (701–440 nm). Considering all the calculated parameters between all the complexes, it was identified that **1B** was the best red, **2B** was the best green, and **3D** was the best blue emitter. Thus, with the mixture of these complexes, a dual host–guest system with **3D**-**1B** and an RGB (red–green–blue) system with **3D**-**2B**-**1B** are proposed, to produce white LECs.

## 1. Introduction

The concept of Solid-State Lighting (SSL) promotes energy savings and greenhouse gas reduction compared to conventional lighting (incandescent bulbs or halogen lamps) [1,2]. The most common SSL devices are LED (Light Emitting Diode) and OLED (Organic Light Emitting Diode) [3,4]. LED technology is based on high purity inorganic semiconductors, for example: AlGaAs, InGaN, GaN, and ZnSe, among others. These materials provide highly efficient and convenient point sources of light of different colors, depending on the semiconductor used [3]. Instead, OLEDs are processed in a multilayer system, using neutral organic or organometallic luminescent compounds, sandwiched between two electrodes [5,6,7]. To improve the manufacturing, costs, and performance of these SSLs, LECs (Light Emitting Electrochemical Cells) emerge as a promising alternative [8].

All SSL systems work through electron–hole recombination processes, where electrons are injected from the cathode into the luminescent material and holes are injected from the anode into the luminescent material. The recombination zone must take place in the middle of the luminescent material, producing an exciton, which if radiatively deactivated could produce light [9,10,11,12]. LEC devices contain a thin film of an ionic luminescent material between a cathode and an anode. This material is usually an ionic transition metal complex (iTMC) [13,14] and their films can be easily processed from an organic solution, using the spin-coating technique, which provides thinner and homogeneous layers [15]. Due to the ionic nature of the iTMC (unlike neutral compounds used in OLEDs), multilayer arrangements are not required in LEC because the electron–hole recombination is determined by the ionic mobility of the active molecules; therefore, LECs are a low cost alternative compare to OLEDs [16].

The cyclometalated Ir(III) complexes ([Ir(C^N)_2_(N^N)]^+^, where C^N is a cyclometalating ligand and N^N is an ancillary ligand) show the best performance as luminescent material in LECs [17,18,19]. The Ir atom is characterized by a high ligand-field splitting energy, thus in their complexes the metal centered (MC) excited states are less thermally accessible, avoiding the non-radiative decays [20,21]. Besides, the spin–orbit coupling (SOC) is increased in Ir-iTMC compared to complexes with metals of the 1st or 2nd transition rows, thus, a very effective singlet-to-triplet intersystem crossing occurs in Ir-iTMC, promoting radiative decay with a high quantum yield [22].

Another outstanding characteristic is that the Ir(III) complexes show emissions in a wide range of the visible spectrum, which can be modulated by the incorporation of an electron donor and/or acceptor substituents in both C^N and N^N ligands [22,23,24]. This can be understood since the electron density distributions of the frontier molecular orbitals (HOMO: Highest Occupied Molecular Orbital, and LUMO: Lowest Unoccupied Molecular Orbital) are similar in most of the Ir-iTMCs [23,25]. The HOMO has contributions from Ir dπ orbitals and π orbitals of the C^N ligand, and the LUMO is composed mainly of π* orbitals centered in the N^N ligand [22,23,26]. Consequently, the emitting triplet state (T_1_) commonly has a ^3^MLCT/^3^LLCT (MLCT: metal-to-ligand charge transfer, LLCT: ligand-to-ligand charge transfer, from C^N to N^N) mixed character [22,23,25,26]. The modulation of the emission energy involves, for example, the use of electron-withdrawing substituents in the C^N ligand, decreasing the electron density on the metal, leading to the stabilization of the HOMO level. Also, electron donor substituents can be incorporated into the N^N ligand, mainly destabilizing the LUMO. The HOMO–LUMO gap is increased with both strategies, obtaining high emission energies. Conversely, with the HOMO destabilization and LUMO stabilization, emissions at lower energies can be obtained [20,25,27].

The design of efficient Ir(III) complexes for LECs involve, in many cases, complicated synthetic procedures of the ligands, obtained at very low yields, affecting the final cost of the devices [19,28,29]. In this sense, the challenge to produce new Ir-iTMCs should involve the use of ancillary and cyclometalating ligands with favorable synthetic routes, providing specific emission colors and, of course, stable compounds.

Achieving full color displays, technologies and versatile lighting applications (lighting in public spaces, highways, advertising, etc.) with SSL devices, is one of the main challenges in the communication, lighting, and computer industries [30,31]. For this target, white light devices with high efficiency and a long lifetime are essential. White light emission from an LEC can be produced with RGB (red–green–blue) emitters, in a triple mix of materials in a single layer, containing the primary colors, since the white output light is a mixture of these three components [32,33,34]. Alternatively, it is possible to use an orange (O) emitter and a blue emitter as complementary colors to produce white light [35]. Since a wide range of colors have been achieved in LECs with Ir-iTMCs, the white light with blends of these complexes has also been explored [36,37,38,39,40]. These systems are called host–guest, where a blue and/or green emitter host is doped with a small concentration of a red emitter guest, in order to promote an incomplete energy transfer from the blue–green complex to excitate a red complex, giving white light from the mixture of the blue–green and red emissions [36,37,38,39,40]. One of the most cited studies that has used this strategy involves the use of [Ir(dfppz)_2_(dedaf)]^+^ (dfppz: 1-(2,4-difluorophenyl)-1*H*-pyrazole, dedaf: 9-diethyl-4,5-diazafluorene) as a blue–green emitter (emission at 491 nm) and [Ir(ppy)_2_(biq)]^+^ (ppy: 2-phenylpyridine, biq: 2,2-biquinoline) as a red emitter (emission at 672 nm), obtaining a white LEC with an external quantum efficiency (EQE) of 4% [36]. Another example involved the use of [Ir(dfppz)_2_(dtb-bpy)]^+^ (dtb-bpy: 4,4′-di-tert-butyl-2,2′-bipyridine) as a blue–green complex (emission at 492 nm) and the same red [Ir(ppy)_2_(biq)]^+^ complex; however, it showed a low efficiency compared to the other case (EQE: 3.2%) [18]. The latter was improved using a mixture of [Ir(dfppz)_2_(dtb-bpy)]^+^ and [Ir(ppy)_2_(biq)]^+^, more an orange complex [Ir(ppy)_2_(dasb)]^+^ (dasb: 4,5-diaza-9,90-spirobifluorene), increasing the EQE up to 6.3% [38].

Considering the wide range of emitting colors obtained from the Ir-iTMCs, the contributions to white light can be feasibly increased, exploring strategic blends of C^N and N^N ligands to produce new stable RGB Ir(III) complexes. According to the antecedents commented on, in this work are evaluated by theoretical calculations, the photophysical properties of an extended family of Ir(III) complexes (see Figure 1) with a mix of C^N and N^N ligands, with different electron-donating and/or -withdrawing natures and different structures to enhance the LEC performance. Therefore, the design of three series was carried out with the purpose of tuning the emitting color of the complexes, from red to blue, mainly determined by the cyclometalating ligand. In this sense, series **1** was expected to show a red emission; series **2**, a green emission; and finally, series **3**, a blue emission. However, some discrepancies were observed, namely, the **3E** complex exhibits strongly green-shifted emissions, which will be discussed in more depth in this study.

The calculations were carried out using the Density Functional Theory (DFT) level, providing a deep structural characterization of the ground and excited states, as well as a study of the absorption and emission properties, and the associated photophysical efficiency parameters, to identify the complexes with the best emitting properties. Finally, the best combinations of blue, green, and red emitters are proposed with the aim to contribute to the strategic design of white LECs.

## 2. Computational Details

All calculations were performed using the DFT approach with the B3LYP functional [41,42], which proved to be the most suitable to reproduce the absorption and emission properties of the [Ir(ppy)_2_(bpy)]^+^ complex, which was used as the reference complex for the DFT benchmark (see Appendix A). The LANL2DZ [43] basis set and quasi-relativistic pseudopotential were adopted for the Iridium atom, while the 6-31G(d,p) [44,45] basis set was used for the other atoms. The optimized structures for all complexes correspond to the energy minima, according to the vibrational frequencies (real values). The Time-Dependent DFT (TD-DFT) methodology was used to calculate the first 50 and 6 triplet excited states, respectively. The triplet (T_1_) excited states were optimized using TD-DFT gradients (TD-DFT optimization). Emission energies were estimated as the vertical energy difference between the total energies of the relaxed triplet state and ground state at the optimized triplet geometry. The continuous solvent effect was incorporated by the IEF-PCM model (polarized continuum model), [46] using dichloromethane as the solvent. The optimized structures of ^3^MC states were also determined; for this purpose unrestricted triplet optimization calculations were performed on the bases of the geometry of the distorted excited triplet state, where the six coordinate bond lengths around the metal atom (Ir-C_C^N_, Ir-N_C^N_ and Ir-N_N^N_) are all gradually elongated (up to about 0.8 Å) and spin density distribution calculations confirm that the metal-centered excited states are obtained, as described in the literature [47,48]. The ionization potentials (IP), the electron affinities (EA), and the hole/electron reorganization energies were obtained by the differences between the total energies of the molecular system in its fundamental state and with one more electron or one less electron [49,50,51]. All the calculations were performed in the Gaussian16 program package [52] and the wavefunction analyses were performed in the Multiwfn 3.4 code [53].

## 3. Results and Discussion

### 3.1. Molecular Geometries in the Ground States and Lowest Excited Triplet States

The different Ir(III) complexes studied are shown in Figure 1, with the numbering of some key atoms. According to this nomenclature, the selected geometric parameters of the ground states (S_0_) and the excited triplet states (T_1_, T_2_ or T_3_) are summarized in Table 1. All the data of the determined geometric parameters are shown in Appendix A.

The calculated data are in agreement with the bond lengths and angles reported in literature for similar cyclometalated Ir(III) complexes [15,19,54,55]. In the ground state it is possible observe that the all complexes have a distorted octahedral geometry around the Ir metallic center, and exhibit similar geometric parameters between them, namely, for the Ir-C and Ir-N bonds of the C^N ligand, an average length of 2.02 and 2.08 Å is found, respectively; while the Ir-N_N^N_ bond lengths are in the range of 2.19 and 2.32 Å. This is significantly longer than those for C^N, which is attributed to the strong *trans*-effect of the C donor in the C^N ligands, in agreement with the reported literature for the Ir(III) complexes with similar characteristics [56,57,58,59]. Furthermore, the bond angles involving the metallic center are also very similar between all complexes; the C_1_-Ir-N_4_ angle is ~96° and the C_1_-Ir-N_3_ angle is around 172–179° for all the complexes. Finally, the C_1_-Ir-C_2_ angle is range between 82–89°.

Comparing the geometric parameters of the triplet excited states with respect to the parameters of the S_0_ structures, marginal variations are found in the distances between the metal center and the ligands; some significant variations were found for Ir-N_N^N_ (Ir-N_2_ and Ir-N_3_) in almost all complexes, where a shortening is identified in the excited states.

### 3.2. Frontier Molecular Orbitals Analysis 

To understand the absorption and emission properties of the studied complexes, it is necessary to analyze their electronic structures in the ground state, namely: the electron distributions of HOMO and LUMO, the energies of the frontier orbitals, and HOMO–LUMO gaps (∆HL). The contributions of molecular fragments to each molecular orbital (including energies) are summarized in Appendix A. Figure 2 depicts the energy diagram showing the HOMO and LUMO surfaces and ∆HL of all complexes.

The HOMO orbital is distributed between the d-orbital of the metal center (33–41%) and the phenyl ring orbitals of the C^N ligand (57–62%), and the LUMO orbital being exclusively on the N^N ligand (~95%) in all complexes. This electron density distribution agrees with the literature data for similar Ir(III) complexes [15,25,55].

Series **1** displays small values of ∆HL; **1A** = 3.00 eV and **1B** = 3.39 eV. With the increase of an aromatic ring fused in the framework of the **A** ancillary ligand, it is incrementing the π-accepting character of this ligand. The increment of conjugation in the ancillary ligands causes an enhancement of the electron withdrawing effect, therefore, the LUMO orbitals are stabilized, reducing the ∆HL of **1A** compared to **1B** [60,61]. Thus, since **1A** shows the lowest ∆HL of the complexes studied, the absorption and emission energy of this complex is expected to be the most red-shifted.

Series **2** has intermediate ∆HL values; **2B** = 3.65 eV and **2C** = 3.61 eV. This variation, with respect to series **1,** is mainly attributed to the energetic stabilization that the HOMO orbital undergoes due to the electron-withdrawing character of the fluorine atoms present in the C^N ligand [62,63,64,65]; therefore, series **2** is expected to present emissions in the green region. This trend has been observed in similar Ir(III) complexes, where the presence of the electron-withdrawing substituents present in the aromatic rings of the C^N ligands promote greater stabilization of the HOMO orbital, demonstrated, for example, by cyclic voltammetry studies, where the oxidation peak associated with the C^N ligand and the d orbitals of the metal is shifted to higher potentials, with respect to the oxidation of the analogous complexes with the C^N ligands without the electron withdrawing substituents [66].

Finally, the largest ∆HL values are for the series **3**; **3B** = 4.02 eV and **3C** = **3D** = 4.01 eV, except for **3E** = 3.51 eV. The introduction of the N onto the aromatic ring of the C^N ligand, plus the presence of the fluorine atoms, has a significant effect on the HOMO energy stabilization, [23] increased in the HOMO–LUMO energy gap compared to the complexes of the series **1** and **2**, which probably will result in blue-shifted emissions. In the case of the **3E** complex, its ∆HL is considerably reduced due to the stabilization of the LUMO by the presence of the **E** ligand, with high electron delocalization that increases their electron acceptor character, as has been described in analogous Ir(III) complexes with a 2,2′-biquinoline as N^N ancillary ligand [38,67].

### 3.3. Absorption Properties

The absorption properties were obtained in dichloromethane on the optimized ground state geometries, the analysis of the absorptions was focused on MLCT bands in the region between 320 to 440 nm and are listed in Table 2.

The lowest absorption bands of series **1** are found between 358 to 437 nm, involving HOMO (HOMO-3) → LUMO (LUMO+1) orbitals. For series **2**, these absorption bands are located at slightly higher energies than series **1**, between 346 and 410 nm, which arises mainly from HOMO (HOMO-1, HOMO-3, HOMO-4) towards LUMO (LUMO+1, LUMO+2) orbitals. Finally, in series **3,** the absorption bands exhibit a considerable blue-shifting (319 to 391 nm) and two trends can be observed, namely, for **3B** and **3C** the lowest lying absorptions are attributed to the HOMO (HOMO-1, HOMO-2) →LUMO (LUMO+1, LUMO+2) transitions, while **3D** and **3E** involves HOMO-1 to HOMO-6 →LUMO orbitals. According to the analysis carried out on the compositions of the molecular orbitals, it is observed that, in all the complexes, the low-energy absorptions have mainly a mixed transition character ^1^MLCT/^1^LLCT/^1^ILCT (ILCT: intraligand charge transfer); in the case of **3E**, this additionally presents a transition with a strong ^1^LC (LC: ligand centered) character, centered on the ancillary ligand (2,2′-biquinoline, **E**).

In summary, we observe that the presence of the fluorine atoms in the C^N ligand (in series **2**), as well as the additional incorporation of the nitrogen atom in the C^N ligand skeleton (series **3**), has a significant influence in the blue-shift of the lowest absorption bands compared to series **1**, which allows the prediction that such complexes present the expected emission at a higher energy. On the other hand, the remarkable participation of MLCT in the region of interest could be indicative of an efficient intersystem crossing throughout the three series, which anticipates a good emission performance.

### 3.4. Emission Properties

The luminescent properties of the complexes in this study were determined based on the minimum energy structures of the lowest triplet excited states, obtained using the TD-DFT formalism. The main results are listed in Table 3.

The obtained emission energies are 1.77 and 2.07 eV for complexes **1A** and **1B**, respectively, indicating that the increase in conjugation of the ancillary ligand **A** leads to a more red-shifted emission, in agreement with the HOMO–LUMO gap. In both complexes, the emitting state (T_1_) has a mixed character of ^3^MLCT/^3^LLCT and arises from the LUMO→HOMO transition, as can be noted in Figure 3 (see Appendix A for all complexes).

In series **2**, it is observed that the fluorine atoms in the cyclometalating ligand cause a displaced emission at high energy in complexes **2B** and **2C** (~2.26 eV), compared to series **1**, as we expected, since this strategy has been widely used to favor blue-shifted emissions [64,65]. For both **2B** and **2C** the emitter state found corresponds to T_2_. The emission of **2B** originates from the LUMO→HOMO-1 transition with a ^3^MLCT and ^3^ILCT character, while, for **2C**, it is predominantly contributed to by the LUMO+1(LUMO+2)→HOMO transitions and its emissive state shows a mixed ^3^MLCT/^3^LLCT/^3^LC character.

In series **3**, the addition of fluorine and nitrogen atoms to the C^N ligand blue-shifts the emission further with respect to series **2**, i.e., the emission energies obtained were 2.76, 2.73 and 2.82 eV, for **3B**, **3C** and **3D**, respectively. The exception is **3E**, which displays a significantly lower emission energy compared to the rest of the complexes, shifting its emission to the yellow region of the visible spectrum; this behavior is attributed to the increased conjugation of the **E** ancillary ligand, which stabilizes the energy of LUMO, as commented previously. The emitting state for **3B**, **3D** and **3E** corresponds to T_2_, while for **3C,** it is T_3_. Two trends are observed in the nature of the emitting excited state: the complexes **3B** and **3D** can be described with ^3^MLCT/^3^LLCT mixed characters and for **3C** and **3E**, a mixed ^3^MLCT/^3^LLCT/^3^LC character. In all these complexes, the deactivation pathway originates from LUMO towards HOMO (HOMO-1, HOMO-2).

Regarding the emission energies found, the designed complexes showed a significant emission color tuning from deep red to blue, i.e., the complexes **1A** and **1B** could be exhibiting red emissions; for **2B**, **2C** and **3E** the emissions ranged at green, and finally, emissions at blue should be displayed for **3B**, **3C** and **3D**.

### 3.5. Phosphorescence Quantum Efficiency

From the photoluminescence quantum yield (*Φ*) we can quantify the efficiency of the emission process. The *Φ* is determined by the competition between the radiative rate constant (*k_r_*) and the non-radiative rate constant (*k_nr_*), according to the following relation: [68,69]
*Φ = k_r_/(k_r_ + k_nr_)*(1)

The *k_r_* is usually expressed as: [63]
*k_r_ = γ(〈Ψ_S1_|H_S0_|Ψ_Tn_〉^2^μ_S1_^2^)/(∆E(S_1_ − T_n_)^2^); with γ = (16π^3^10^6^n^3^E_emi_^3^)/(3hε_o_)*(2)
where *〈**Ψ**_S_*_1_*|H_S_*_0_*|**Ψ**_Tn_**〉* is the spin–orbit coupling (SOC) matrix element between the S_1_ state and the emitting state (T_1_, T_2_ or T_3_), *μ**_S_*_1_ is the transition electronic dipole moment in S_0_→S_1_ transition, *∆E(S*_1_
*− T_n_*) is the energy gap between the S_1_ state and the emitting triplet state, and *n*, *E_emi_*, *h* and *ε*_0_ are the refractive index of the medium, emission energy, Planck’s constant, and the permittivity in vacuum, respectively. This expression is applicable to coordination compounds with a heavy metal center since the radiative rate is directly proportional to the SOC matrix element related to the emitting triplet and singlet states, and inversely proportional to the degree of mix between them (*∆E*(*S*_1_ − *T_n_*)). In these type of complexes, large intersystem crossing rates (ISC) predominate, so the fluorescence rate can be considered null, as well as some non-radiative pathways (inverse ISC, internal conversion, conversion from MLCT to MC) [70].

The SOC effects can be elucidated from the metal contribution in the emitting state (%^3^MLCT), which were calculated from the sum of the ^3^MLCT contributions from each monoexcitation (according to their respective orbitals involved), considering the corresponding configuration coefficient, as has been reported in the literature [71,72]. The results are summarized in Table 4.

According to Equation (2), higher values of %^3^MLCT and *μ**_S_*_1_ increase the *k_r_* value and, conversely, higher values of *∆E(S*_1_
*− T_n_)* decrease the *k_r_*. We observe that, in all complexes, the %^3^MLCT is high enough, therefore, *k_r_* is favored in all cases. However, the higher values are obtained for the complexes of series **1** and series **3** (except for **3E**), which suggests that the %^3^MLCT is affected by the different C^N ligands used. Interestingly, these complexes showed structural changes in the excited state (T_1_, T_2_ or T_3_) that would favor the metal–ligand interaction, namely, a shortening of the length of Ir-N_N^N_ ligand bonds (Ir-N_2_ and Ir-N_3_).

For *μ**_S_*_1_, the variation of the C^N and N^N ligands do not show a clear trend. The largest values are shown by **1B**, **2B**, **2C,** and **3C** (between 1.26 to 1.00 D), so higher *k_r_* values are expected in these complexes.

On the other hand, a minimal difference between the S_1_ and the emitting state, Δ*E(S*_1_*-T_n_)*, is favorable for enhancing the intersystem crossing (ISC) rate, implying an increase in *k_r_*. The results show significantly lower values for complexes **3D**, **3B**, **1A,** and **1B**, and precisely, in these complexes, the highest values of %^3^MLCT (~27% to 33%) are observed, as expected since a high SOC (%^3^MLTC) promotes an effective ISC.

In summary, considering all the parameters that determine the *k_r_* value, it is observed that the **1A**, **1B**, **2B**, **2C**, **3B,** and **3D** complexes would present characteristics that favor the radiative deactivation processes.

Equation (1) shows that another determining factor in the efficiency of the emission process corresponds to the *k_nr_*. The population of metal-centered (^3^MC, d-d*) triplet excited states is one of the most important deactivation pathways of the phosphorescence, namely, the ^3^MLCT (π-π*) excited state can be rapidly converted to a short-lived ^3^MC (d-d*) state, from which no emission occurs; to avoid this situation, a splitting in energy is required between the ^3^MC and ^3^MLCT states [25,73,74].

The metal–ligand bond lengths and bond angles in ^3^MC states are listed in Appendix A (see Appendix A), along with the spin density distributions determined for the ^3^MC state (see Appendix A) where the metal-centered character is observed. The results show that, in almost all complexes, the main structural change in the ^3^MC states correspond to the distance between the metal and the N atoms located in the axial position (Ir-N_1_/Ir-N_4_), which increase up to 0.6 Å, with respect to ground state. However, complexes **1A** and **3C** showed a relatively small structural distortion between the ^3^MC and S_0_ states, which would explain the lower energies obtained for these ^3^MC states.

In general, it is observed that, in almost all complexes, the relative energy position of the ^3^MC state lies above the emitting triplet state ^3^MLCT, except for complexes **3B** and **3C**, as shown in Figure 4. This is explained due to the electron withdrawing character of the cyclometalating ligand (**3**), therefore, it strongly influences the increase in the energy of the ^3^MLCT excited state, which, in some cases, causes it to exceed the energy of the ^3^MC state. High energies of the ^3^MLCT excited states have been identified for analogous Ir(III) complexes with this cyclometalating ligand [75,76,77,78]. This effect is almost experienced by **3D** (^3^MLCT energy close to ^3^MC energy) and, in the case of **3E**, the effect is not observed since the **E** ligand (electron withdrawing by resonance) compensates for the effect of the cyclometalating ligand.

With regard to the behavior of the **3B** and **3D** complexes, a similar tendency should be expected, since the aliphatic bonds and substituents in the 4 and 4′ positions should not influence the electronic properties. However, to elucidate why, in one case, the ^3^MC (**3B**) is stabilized and in the other, the ^3^MLCT (**3D**) is destabilized, we calculated the root-mean-square deviation (RMSD), determining the difference between two sets of the coordinate values, in this case between the ^3^MLCT and ^3^MC excited states (see Appendix A). For the **3B** complex, a higher value (0.668 Å) was found compared to **3D** (0.494 Å), which would explain the higher energy gap in **3B**, probably due to the high mobility of the substituents in the 4,4′ positions, yielding a ^3^MC state that is more stabilized. Whereas in **3D,** the lower value of RMSD indicates less distortion (in fact, both states have very similar energies), where the ^3^MLCT state is most stable.

The computed adiabatic energy differences between the ^3^MC and ^3^MLCT (∆E(^3^MC-^3^MLCT)) are 0.13, 0.41, 0.26, 0.19, −0.18, −0.26, 0.05, and 0.48 eV for **1A**, **1B**, **2B**, **2C**, **3B**, **3C**, **3D**, and **3E**, respectively. High ∆E(^3^MC-^3^MLCT) values of iTMCs, in the range of 0.26–0.60 eV, have shown high electroluminescent performance in LEC devices [79]. Consequently, the complexes **1B**, **2B,** and **3E** would lead to a lower *k_nr_* and would present a high performance in their application in LEC. While the probability of populating the ^3^MC states increases in complexes **1A**, **2C,** and **3D**, which would result in a more favored *k_nr_*, as the emission of **3B** and **3C** comes from very high ^3^MLCT states (blue-shifted emission), [21] the conversion from the emitting state to the ^3^MC state is easily completed, thereby increasing the probability of non-radiative decay to the ground state and consequently, a significantly larger *k_nr_* is obtained. Note that the **3D** blue emitting state also displays a high ^3^MLCT state.

### 3.6. Charge Transfer Properties: IP and EA

The good performance of LEC devices is determined by the appropriate injection and transfer of holes and electrons, which can be evaluated by the ionization potential (*IP*), electron affinity (*EA*), and reorganization energy (*λ*). The *IP* and *EA* are obtained as: [49,50,51]
*IP = E_N-1_ – E_N_*(3)
*EA = E_N_ – E_N+1_*(4)
where *E_N_*, *E_N+_*_1_, *E_N-_*_1_ are the total energies of the molecular system in its ground state, with one electron more and one electron less, respectively. Small *IP* values indicate easy hole injection from the anode to the HOMO of the iTMC, and large *EA* values can be related to a favored electron injection from the cathode to the LUMO of the iTMC [25,80,81]. As shown in Table 5, the *IP* values gradually increase in the following order: Series **1** < Series **2** < Series **3**, which is consistent with the HOMO energy levels (see Section 3.2). Therefore, the C^N ligand largely determines the HOMO level, establishing that ligand **1** causes a lower hole injection energy barrier compared to ligands **2** and **3**. In this sense, series **1** could display the best hole injection performance.

Related to the *EA*, the highest values are found in **1A** and **3E**, showing agreement with the lowest energies of the LUMO levels, according to the electron acceptor character of the N^N ligand. Consequently, these three complexes will have an enhanced electron injection ability compared to other complexes.

On the other hand, the reorganization energy can be used to estimate the charge transport rate and balance between holes (*λ**_h_*) and electrons (*λ**_e_*) according to the following expression: [23,82,83]
*λ_h_ = IP – HEP*(5)
*λ_e_ = EEP – EA*(6)

*HEP* (*EEP*) is the hole (electron) extraction potential and is determined as the vertical energy difference between the ground state and the relaxed state with one electron less (more), using the geometry with one electron less (more), respectively. Generally, a low reorganization energy (*λ**_h_*, *λ**_e_*) is necessary for an efficient charge transport process and in this respect, the full series **3** appears to be more efficient in the hole transport (*λ**_h_* = 0.12 to 0.13 eV) while complexes **1A**, **2C**, **3C,** and **3E** showed greater efficiency in electron transport. However, in all complexes, the hole transport performance is favored over the electron transport ability, due to the higher values obtained of *λ**_e_* (0.32 to 0.46 eV), with respect to *λ**_h_* (0.12 to 0.19 eV).

In addition, we determined the difference between *λ**_h_* and *λ**_e_*, and it was observed that the complexes **1A**, **2C**, **3C,** and **3E** showed less discrepancy (*∆**λ*
*<* 0.21), indicating that the balance of electron and hole transfer could be easily achieved in the emitting layer of the LEC devices.

### 3.7. RGB Systems to Produce White LECs

According to the emission energies determined for each complex studied, in Table 6, in the first line, the complexes were organized from the reddest emitter to the bluest emitter, from left to right, respectively. It can be observed that this order differs with respect to the original tendency designed, since it was expected that the type of cyclometalating ligand chosen was the main factor determining the emission energy. In this sense, the **3E** complex is out of the original tendency since their ancillary ligands promote a strong stabilization of LUMO, which shifted their emissions at lower energies. Then, in the following lines of the table, an arbitrary qualification has been used for each studied complex with respect to the values of *k_r_*, *k_nr_*, *IP*, *EA*, and *Δλ*. This qualification arises from the parameters determined in Section 3.5 and Section 3.6, based on the minimum and maximum values determined, and also considering key values from the literature. Then, the +++ symbol is assigned to a very favourable value, ++ to a favourable value, and + is slightly favourable. In the case of *k_nr_*, a 0 value is added to qualify the parameter with an unfavourable value. The extension of the criteria assigned can be observed in the Appendix A.

By the analysis of *k_r_* and *k_nr_* kinetic parameters, that describe the efficiency of the intrinsic radiative process of the complex, it is observed that, for red emitter complexes, the best combination of these parameters is obtained in complex **1B**, therefore, this is the best candidate to be a good red emitter in an RGB system. Then, if *IP*, *EA*, and *Δλ* parameters are considered, **1A** could be an appropriate red emitter. Next, in terms of the evaluation for the green emitter complexes, the best performance in the kinetic parameters of the phosphorescent processes is found for **2B**, therefore being the best candidate as a green emitter. Then, if the charge transport and injection parameters are considered, plus its balance, **3E** and finally **2C** could be proposed.

Finally, for the emitter complexes in the blue range, only **3D** could act as an appropriate emitter for an RGB system, since the other blue complexes have a very unfavourable *k_nr_*. This behaviour in **3C** and **3B** is due to the higher energy of the ^3^MLCT states with respect to ^3^MC, promoting this ^3^MC state as the phosphorescence deactivator, which is known to prefer non-radiative paths.

Consequently, according to this analysis, a proposal for dual host–guest system with blue and red emitters to produce white LECs could be obtained by mixing **3D**–**1B**, as well as **3D**–**1A**. In the case of an RGB system, the proposal would be to assemble the **3D**–**2B**–**1B** complexes.

## 4. Conclusions

A detailed investigation is reported on the geometrical and electronic structures, emission properties, charge injection/transport abilities, and phosphorescence efficiency of eight new Ir(III) complexes (classified into 3 series) using DFT and TD-DFT methods. The designed complexes showed a significant emission color tuning from deep red to blue, with emissions ranging from 440 to 701 nm. The results showed that the Ir-N^N ligand bond lengths are shortened for complexes of series **1** and **3***,* which has a direct impact on the metal–ligand interaction, leading to more involvement of the metal in the triplet excited state (%^3^MLCT). The analysis of quantum efficiency showed that the **1A**, **1B**, **2B**, **2C**, **3B,** and **3D** complexes would present characteristics that favor the radiative deactivation processes, while **1B**, **2B,** and **3E** complexes would lead to a lower *k_nr_* and would present a high performance in their application in LEC. In relation to transport and injection parameters, the complexes with the best balance between hole and electron injection are **1A**, **2C**, **3C**, and **3E** (*∆**λ* < 0.21), however, the full Series **3** stands out for being more efficient in the hole transport, and **1A**, **2C**, **3C,** and **3E**, for presenting greater efficiency in the transport of electrons. Finally, we have proposed a host–guest dual system using a mix of **3D** and **1B**, and also an RGB system based on **3D**–**2B**–**1B,** to produce white LECs.

With this family of complexes, corresponding to newly designed structures based on a mix of C^N and N^N ligands, commercially available and/or easy to synthesize, an important contribution of new Ir-iTMCs is provided from a theoretical approach, which is totally experimentally viable.

## Figures and Tables

**Figure 1 molecules-27-02623-f001:**
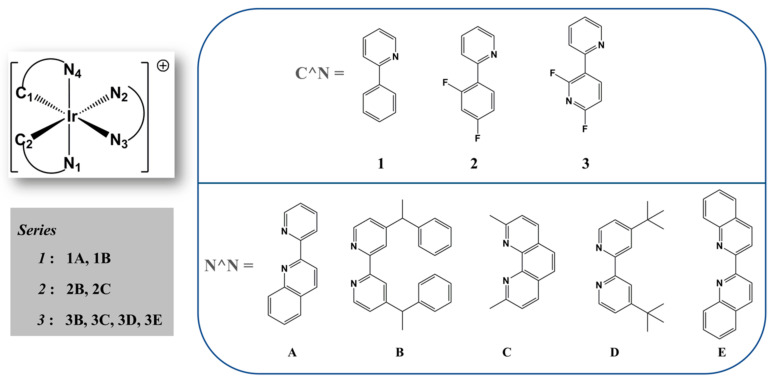
Molecular structures of the Ir(III) complexes studied. Labeling of the atoms coordinated with the metallic center is included to guide the description of the geometric parameters.

**Figure 2 molecules-27-02623-f002:**
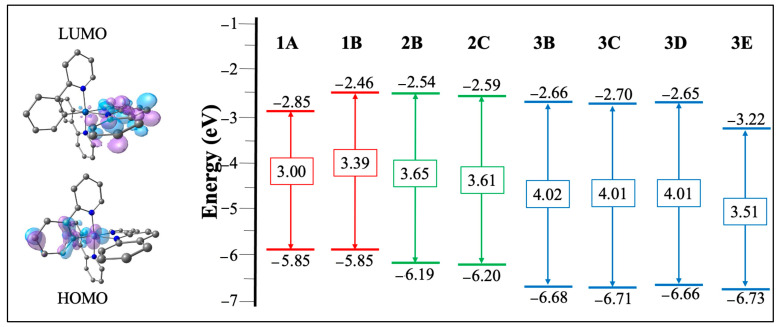
Molecular orbitals diagram for all complexes and HOMO–LUMO plots for **1A** as representative of all complexes (for the rest of the complexes, see Appendix A).

**Figure 3 molecules-27-02623-f003:**
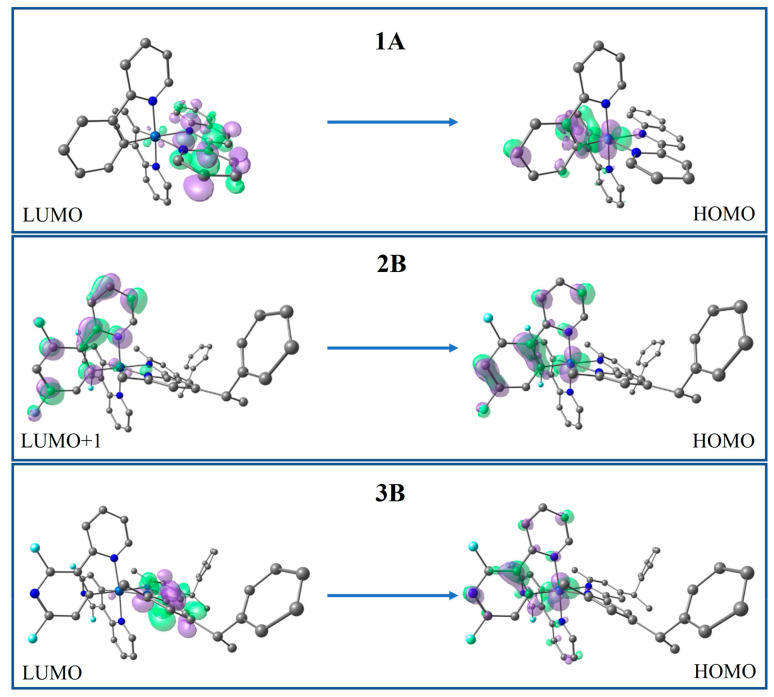
Radiative deactivation pathway of the triplet excited state of **1A**, **2B** and **3B**, as representative of each series (for the rest of the complexes, see Appendix A).

**Figure 4 molecules-27-02623-f004:**
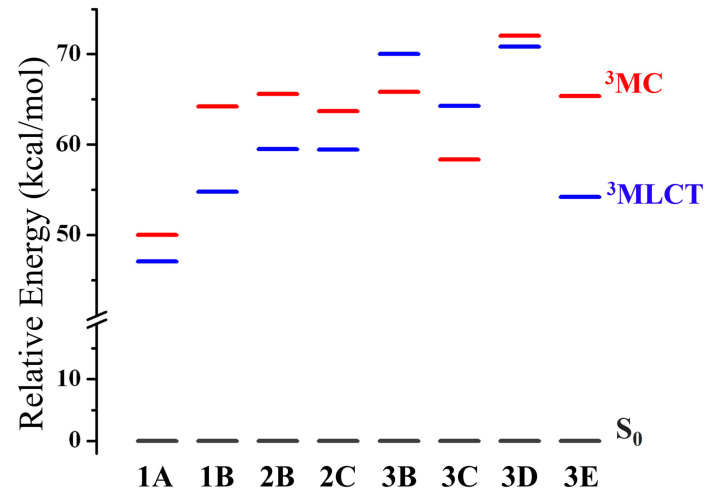
Energy level diagram of all complexes under study of ^3^MLCT, ^3^MC and S_0_ states (normalized).

**Table 1 molecules-27-02623-t001:** Selected optimized geometric parameters of all complexes under study in the S_0_ and triplet excited states (T_1_, T_2_ or T_3_) determined at the B3LYP/6-31G(d)-LANL2DZ level of theory.

	1A		1B		2B		2C	
	S_0_	T_1_	S_0_	T_1_	S_0_	T_2_	S_0_	T_2_
Bond length (Å)								
Ir-C_1_	2.02	2.02	2.02	1.99	2.02	2.01	2.02	2.02
Ir-C_2_	2.03	1.98	2.02	2.01	2.02	2.02	2.02	2.01
Ir-N_1_	2.08	2.08	2.07	2.07	2.07	2.04	2.09	2.10
Ir-N_2_	2.20	2.23	2.31	2.24	2.29	2.31	2.28	2.29
Ir-N_3_	2.32	2.24	2.31	2.27	2.30	2.30	2.29	2.31
Ir-N_4_	2.08	2.09	2.08	2.09	2.08	2.09	2.07	2.04
**Bond angle (deg)**								
C_1_-Ir-N_4_	96.2	97.9	95.7	96.9	95.5	94.9	95.8	95.9
C_1_-Ir-N_3_	169.4	164.4	177.1	175.9	177.5	177.2	172.2	172.6
C_1_-Ir-C_2_	85.4	90.2	82.4	88.2	82.2	83.2	82.3	83.1
	**3B**		**3C**		**3D**		**3E**	
	**S_0_**	**T_2_**	**S_0_**	**T_3_**	**S_0_**	**T_2_**	**S_0_**	**T_2_**
**Bond length (Å)**								
Ir-C_1_	2.02	2.00	2.01	2.02	2.01	2.00	2.01	2.02
Ir-C_2_	2.01	2.02	2.02	2.02	2.01	2.00	2.02	2.02
Ir-N_1_	2.08	2.06	2.09	2.08	2.08	2.08	2.08	2.07
Ir-N_2_	2.29	2.20	2.28	2.23	2.19	2.18	2.29	2.22
Ir-N_3_	2.28	2.24	2.28	2.23	2.19	2.18	2.28	2.22
Ir-N_4_	2.08	2.10	2.09	2.09	2.08	2.08	2.09	2.10
**Bond angle (deg)**								
C_1_-Ir-N_4_	95.7	96.5	95.6	95.5	95.5	96.9	95.5	94.9
C_1_-Ir-N_3_	173.5	177.7	172.6	177.2	172.5	170.1	178.9	178.3
C_1_-Ir-C_2_	81.7	85.0	81.7	82.6	89.0	94.5	82.1	83.9

**Table 2 molecules-27-02623-t002:** Absorption properties calculated from TD-DFT approach, in dichloromethane as solvent. Determined at the B3LYP/6-31G(d)/LANL2DZ level of theory.

System	State	E_abs_ (λ_abs_)	*f*	Monoexcitations	Description
**1A**	S_4_	2.83 (437)	0.077	H-3→L (80%)	Ir(d) + C^N(π)→N^N(π*); ^1^MLCT/^1^LLCT
	S_5_	2.99 (414)	0.042	H→L + 1 (97%)	Ir(d) + C^N(π)→C^N(π*); ^1^MLCT/^1^ILCT
**1B**	S_2_	3.14 (394)	0.036	H→L + 1 (96%)	Ir(d) + C^N(π)→C^N(π*); ^1^MLCT/^1^ILCT
	S_6_	3.46 (358)	0.064	H-3→L (87%)	Ir(d) + C^N(π)→N^N(π*); ^1^MLCT/^1^LLCT
**2B**	S_2_	3.34 (371)	0.035	H→L + 1 (94%)	Ir(d) + C^N(π)→C^N(π*); ^1^MLCT/^1^ILCT
	S_6_	3.65 (440)	0.050	H-3→L (54%) H-4→L (18%)	Ir(d) + C^N(π)→N^N(π*); ^1^MLCT/^1^LLCT Ir(d) + C^N(π)→N^N(π*); ^1^MLCT/^1^LLCT
**2C**	S_3_	3.41 (363)	0.026	H→L + 2 (82%)	Ir(d) + C^N(π)→C^N(π*); ^1^MLCT/^1^ILCT
	S_7_	3.58 (346)	0.050	H-3→L (66%)	Ir(d) + C^N(π)→N^N(π*); ^1^MLCT/^1^LLCT
**3B**	S_2_	3.66 (339)	0.023	H→L + 1 (81%)	Ir(d) + C^N(π)→C^N(π*); ^1^MLCT/^1^ILCT
	S_4_	3.70 (335)	0.035	H-1→L (78%)	C^N(π)→N^N(π*); ^1^LLCT
**3C**	S_3_	3.63 (341)	0.041	H-2→L (67%)	Ir(d) + C^N(π) + N^N(π)→N^N(π*); ^1^MLCT/^1^LLCT/^1^ILCT
	S_7_	3.79 (327)	0.028	H→L + 2 (35%) H→L + 1 (24%) H-1→L (14%)	Ir(d) + C^N(π)→C^N(π*) + N^N(π*); ^1^MLCT/^1^LLCT/^1^ILCT Ir(d) + C^N(π)→N^N(π*); ^1^MLCT/^1^LLCT C^N(π)→N^N(π*); ^1^LLCT
**3D**	S_3_	3.66 (338)	0.054	H-1→L (88%)	C^N(π)→N^N(π*); ^1^LLCT
	S_7_	3.88 (319)	0.092	H-4→L (85%)	Ir(d) + C^N(π) + N^N(π)→N^N(π*); ^1^MLCT/^1^LLCT/^1^ILCT
**3E**	S_3_	3.16 (391)	0.048	H-2→L (45%) H-1→L (35%)	Ir(d) + C^N(π) + N^N(π)→N^N(π*); ^1^MLCT/^1^LLCT/^1^ILCT C^N(π)→N^N(π*); ^1^LLCT
	S_7_	3.60 (344)	0.048	H-5→L (48%) H-6→L (42%)	Ir(d) + N^N(π)→N^N(π*); ^1^MLCT/^1^LC N^N(π)→N^N(π*); ^1^LC

**Table 3 molecules-27-02623-t003:** Excited states properties of the Ir(III) complexes studied calculated from TD-DFT approach.

Complexes	State	λ_emi_/nm (E_emi_/eV)	Main Configuration	Character
**1A**	T_1_	701 (1.77)	L → H (99%)	N^N(π*)→ Ir(d) + C^N(π); ^3^MLCT/^3^LLCT
**1B**	T_1_	599 (2.07)	L → H (97%)	N^N(π*)→ Ir(d) + C^N(π); ^3^MLCT/^3^LLCT
**2B**	T_2_	547 (2.26)	L + 1 → H (69%)	C^N(π*)→ Ir(d) + C^N(π); ^3^MLCT/^3^ILCT
**2C**	T_2_	548 (2.26)	L + 1 → H (48%)	N^N(π*)+ C^N(π*) → Ir(d) + C^N(π); ^3^MLCT/^3^LLCT/^3^LC
			L + 2 → H (24%)	N^N(π*) + C^N(π*) → Ir(d) + C^N(π); ^3^MLCT/^3^LLCT/^3^LC
**3B**	T_2_	448 (2.76)	L → H (79%)	N^N(π*) → Ir(d) + C^N(π); ^3^MLCT/^3^LLCT
**3C**	T_3_	454 (2.73)	L → H-1 (43%)	N^N(π*) → Ir(d) + N^N(π) + C^N(π); ^3^MLCT/^3^LLCT/^3^LC
			L → H (41%)	N^N(π*) → Ir(d) + C^N(π); ^3^MLCT/^3^LLCT
**3D**	T_2_	440 (2.82)	L → H (92%)	N^N(π*) → Ir(d) + C^N(π); ^3^MLCT/^3^LLCT
**3E**	T_2_	574 (2.16)	L → H-2 (26%)	N^N(π*) → Ir(d) + C^N(π); ^3^MLCT/^3^LLCT
			L → H (19%)	N^N(π*) → Ir(d) + C^N(π); ^3^MLCT/^3^LLCT
			L → H-1 (18%)	N^N(π*) → Ir(d) + N^N(π) + C^N(π); ^3^MLCT/^3^LLCT/^3^LC

**Table 4 molecules-27-02623-t004:** Metal–ligand charge transfer character (^3^MLCT, %), transition electric dipole moment *(**μ_S_*_1_, D) and energy gaps between the S_1_ and T_n_ states (*∆E(S*_1_ − *T_n_*), eV) of studied complexes.

Complexes	%^3^MLCT	μ_S1_	*Δ**E(S*_1_ − *T_n_)*
**1A**	32.4	0.61	0.043
**1B**	33.2	1.26	0.057
**2B**	19.6	1.22	0.110
**2C**	20.7	1.11	0.144
**3B**	27.6	0.60	0.020
**3C**	23.4	1.00	0.330
**3D**	30.8	0.14	0.009
**3E**	18.2	0.94	0.275

**Table 5 molecules-27-02623-t005:** The Ionization potential (*IP*, eV), electron affinities (*EA*, eV), hole/electron reorganization energy (*λ*_h_/*λ*_e_, eV) and *∆**λ* (eV).

Complexes	*IP*	*EA*	*λ* * _h_ *	*λ* * _e_ *	∆*λ* = *λ**_e_* − *λ**_h_*
**1A**	5.85	2.83	0.19	0.32	0.13
**1B**	5.85	2.42	0.18	0.45	0.27
**2B**	6.20	2.50	0.16	0.46	0.30
**2C**	6.21	2.55	0.15	0.33	0.18
**3B**	6.70	2.63	0.12	0.44	0.32
**3C**	6.72	2.65	0.12	0.33	0.21
**3D**	6.67	2.59	0.13	0.40	0.27
**3E**	6.75	3.21	0.12	0.32	0.20

**Table 6 molecules-27-02623-t006:** Analyzed photophysics and charge transport parameters to determine the best RGB systems.

Complexes	1A	1B	3E	2C	2B	3C	3B	3D
Color	Red	Red	Green	Green	Green	Blue	Blue	Blue
** *λ_em_/nm* **	701	599	574	548	547	454	448	440
** *k_r_* **	++	+++	+	++	++	+	+++	++
** *k_nr_* **	+	++	++	+	++	0	0	+
** *IP* **	+++	+++	+	++	++	+	+	+
** *EA* **	+++	+	+++	+	+	++	++	++
** *Δλ* **	+++	++	+++	+++	+	+++	+	++

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
