# Peer review of "Theoretical Approach for the Luminescent Properties of Ir(III) Complexes to Produce Red–Green–Blue LEC Devices"

_molecules, 2022, doi:10.3390/molecules27092623_

Round 1

Author Response

Referee 1:

1°) The difference between LED ad LEC, which largely resides on the ionic (or not) nature of the emitting layer) might be somewhat emphazised (made clearer) for the readers in the introduction. Furthermore, in the context of the Ir(III) complexes presently studied, the advantage of the latter technique (only one active layer for emission) is more challenging to obtain with blends of severl phosphors in dual or RGB systems, depending on how they are made and how the energy transfers between nearby complexes. Maybe a word of caution would be warranted regarding this aspect which is presently not addressed in the section of the paper when dual or RGHB systems are discussed based on these compounds.

We appreciate the comments of the reviewer and based on them we have added additional information in the introduction to clarify the difference between an LED and an LEC, and additionally, more arguments have been provided to better understand the operation of host-guest dual or RGB systems.

The changes have been considered in the new version of the article (see highlighted sentences in yellow in the introduction section).

2°) The assumptions under which eq. (2) has been derived should be recalled in the discussion. For instance, I think this expression assumes no non-radiative (reverse isc/internal conversion) relaxation occuring directly from the triplet state. Furthermore I think it considers k(isc) >>> k(fluorescence) (or k(fluorescence) = 0) and, among the nonradiative processes, neglects any internal conversion from 1MLCT state to the 1MC state and the competitive isc process from the 1MLCT state to the 3MC state. Am I correct ?

The reviewer is right, in this sense we are clarified all the assumptions involved in equation2 in the new version of manuscript (see added information highlighted in yellow in the 3.5 section).

Typos:

1°) L. 61 : « understood » rather than « understanding »

2°) Line 189: «with result in» rather than «will conduce to»

3°) Line 212: Is it really «3LC» instead of «1LC». If yes, explain why. Explain also for which ligand (diimine or arylpyridine) is this excited state character given

4°) Line 300: « …a splitting in energy… » instead of «…a splitting  energy… ».

5°) Line 368: «chosen » instead of «choose» and «…factor determining… » instead of « …factor to

determine… »

6°) Line 370: «following » instead of «followed»

7°) Line 410: «In relation to » instead of «Respect»

The 1°) - 7°) changes indicated has been considered in the new version of the article (see highlighted sentences in yellow).

Reviewer 2 Report

This article is focused on the conduction of DFT evaluation of eight iTMC and evaluation of their photophysical properties. Although the conceptualization and organization of paper is relevant to many of the literature precedents, I still think this paper should be suitable for publication in Molecules, after conducting the revision recommendations as listed below:

  1. I think the design of N^N chelate “B” is unnecessary because the benzyl substituents are not going to provide anything extraordinary. They should switch to the N^N chelate “D” instead. All photophysical results will be more-or-less unchanged.
  2. On page 4, the authors stated that Ir-N distances of N^N chelate are significantly longer than the Ir-N distance of C^N chelates. Please provide an explanation for this observation.
  3. In Figure 4, the energy of MC state of 3B and 3C is lower than that of MLCT excited state. Please provide an explanation for this observation. Moreover, the electronic effect of N^N chelates “B” and “D” should be nearly identical. Why is this deduction cannot be confirmed in this energy diagram? Are there any experimental precedents to justify this claim?

Author Response

Referee 2:

This article is focused on the conduction of DFT evaluation of eight iTMC and evaluation of their photophysical properties. Although the conceptualization and organization of paper is relevant to many of the literature precedents, I still think this paper should be suitable for publication in Molecules, after conducting the revision recommendations as listed below:

  1. I think the design of N^N chelate “B” is unnecessary because the benzyl substituents are not going to provide anything extraordinary. They should switch to the N^N chelate “D” instead. All photophysical results will be more-or-less unchanged.

We value the reviewer's comment and understand that in the electronic properties, in solution, there should be no great differences between complexes containing B or D as auxiliary ligand, keeping the ligand cyclometalated. However, it is known that both in the evaluation of the photophysical properties in the solid state, and in the evaluation of the iTMC in LEC, the structure of the ligands is determining, namely, the bulky ligands could lead to better performance in LEC, as has been previously studied in articles from our group and from other groups also focused on LEC devices (10.1039/C6NJ00213G; 10.1039/c3dt52067f; 10.1002/adfm.201000043; 10.1002/adfm.200801767)

In order to clarify the use of these two different ancillary ligands (B and D), we incorporated a brief explanation in the new version of the manuscript (see highlighted sentences in green in the introduction section).

  1. On page 4, the authors stated that Ir-N distances of N^N chelate are significantly longer than the Ir-N distance of C^N chelates. Please provide an explanation for this observation.

We are considered this observation of the new version of the manuscript, and we have included an explanation (see sentence highlighted in green in section 3.1).

  1. In Figure 4, the energy of MC state of 3B and 3C is lower than that of MLCT excited state. Please provide an explanation for this observation.

This effect is mainly due to the electron withdrawing character of the cyclometallating ligand (2′,4′-difluoro-2,3′-bipyridinato-N,C4′), therefore, it strongly influences the increase in the energy of the 3MLCT excited state, which in some cases can cause it to exceed the energy of the 3MC state. High energies of the 3MLCT excited states have been identified for analogous Ir(III) complexes with this cyclometallating ligand (10.1021/cm3010453; 10.1039/C5CP05328E; 10.1016/j.jphotochem.2017.09.061; 10.1016/j.jphotochemrev.2016.10.001; 10.1021/ic801643p). Additionally, this effect can also be observed in 3D where its 3MLCT state has a high energy (3MLCT energy close to 3MC energy), conversely, 3E is not affected because the presence of E ligand, which is an electron withdrawing by resonance, compensates the high electron withdrawing effect of ligand 3 lowering the energy of the 3MLCT excited state.

Moreover, the electronic effect of N^N chelates “B” and “D” should be nearly identical. Why is this deduction cannot be confirmed in this energy diagram? Are there any experimental precedents to justify this claim?

We have understood that the electronic properties of the 3B and 3D complexes must be very similar, in fact, many of the calculated photophysical parameters demonstrate this. However, in order to understand why the 3MC and 3MLCT excited states are stabilized for 3B and 3D, respectively, we perform calculations of root-mean-square deviation (RMSD), that determine the difference between two sets of the coordinate values, in this case between the 3MLCT and 3MC excited states. For the 3B complex, a higher value (0.668 Å) was found compared to 3D (0.494 Å), which would explain the higher energy gap in 3B, probably due to the high mobility of the substituents in the 4,4' positions, yielding a 3MC state more stabilized. Whereas in 3D the lower value of RMSD indicates of less distortion (in fact both states have very similar energies), where the 3MLCT state is most stable. To address this comment, we incorporated a probable explanation of this behavior in the new version of the manuscript (see highlighted sentences in green in the section 3.5).

Reviewer 3 Report

The manuscript reported a theoretical approach to study the luminescent properties of Ir(III) complexes, where a mixture of cyclometalating and ancillar ligands are taken into account. The emitting capability are calculated by using DFT and TD-DFT calculations. In particular, the HOMO-LUMO band gaps, the phosphorescence quantum efficiency and the charge transfer properties are obtained and outcomes for different complexes are compared. The topic of the manuscript is certainly of interest for the Molecules readers, but major revision are necessary before an eventual publication.

The crucial points are:

  1. The choice of the series and the ligands is good, but there are several lack complexes. Why are not present 1C, 1D, 1E, 2A, 2D, 2E and 3A? Without complete data, where only a ligand (N or C) is changed, it this difficult to fully rationalize the role of each ligand.  
  2. The role of C and N ligands on the HOMO and LUMO stabilities are clearly reported in Figure 2. Authors should better explained the reason of this behavior. For example, 2 and 3 strongly stabilized the HOMOs, while the LUMOs are less affected, while B seems poorly affects the LUMOs in the three series. A rational explanation of these trends is necessary.
  3. In some systems, the 3MC state is more stable than the 3MLCT one. If, as describe in the text “optimized structures of 3MC states were also determined, considering a triplet distorted geometry by a gradual elongation of the metal-ligand bond lengths as described in the literature.”, authors should better explain because this inversion of stability is present only in some systems and should correlate with ligands.

Minor revision

3MC optimized geometry is crucial to determine the properties of the different complexes. I suggest to explicitly add in the computational details how these geometries are obtained and not limited this explanation to a couple of references.

Author Response

Referee 3:

The manuscript reported a theoretical approach to study the luminescent properties of Ir(III) complexes, where a mixture of cyclometalating and ancillar ligands are taken into account. The emitting capability are calculated by using DFT and TD-DFT calculations. In particular, the HOMO-LUMO band gaps, the phosphorescence quantum efficiency and the charge transfer properties are obtained and outcomes for different complexes are compared. The topic of the manuscript is certainly of interest for the Molecules readers, but major revision are necessary before an eventual publication.

The crucial points are:

  1. The choice of the series and the ligands is good, but there are several lack complexes. Why are not present 1C, 1D, 1E, 2A, 2D, 2E and 3A? Without complete data, where only a ligand (N or C) is changed, it this difficult to fully rationalize the role of each ligand.

As mentioned in the manuscript, in the cyclometalated Ir(III) complexes proposed the mix of some cyclometalating ligands with some ancillary ligands is carried out in order to obtain red, green and blue emitters. These colours are obtained mainly as a function of the HOMO energy, that is, as a function of the cyclometalating ligand nature, so red emitters are expected for series 1, green for series 2 and blue for series 3. Not all the ligands were mixture, because the objective is to propose new Ir(III) complexes (unpublished), that can easily synthetized, and use in the next studied of our group. The complexes mentioned by the reviewer have been published, namely: 10.1080/15421406.2013.849444 (1C); 10.1021/acsami.8b08176 (1D); 10.1002/cplu.201800198 (1E); 10.1021/om900691r (2A and 2E); 10.1002/chem.201300457 and 10.1002/9781119007166.ch7 (2D).

The exception is complex 3A, which is not published, but was not calculated since the emission energy is expected to be very close to the emission energy of complex 3E.

  1. The role of C and N ligands on the HOMO and LUMO stabilities are clearly reported in Figure 2. Authors should better explained the reason of this behavior. For example, 2 and 3 strongly stabilized the HOMOs, while the LUMOs are less affected, while B seems poorly affects the LUMOs in the three series. A rational explanation of these trends is necessary.

In the new version of the manuscript, we added some information to complement the tendencies of the HOMO and LUMO orbitals (see highlighted in cyan in the section 3.2).

Regard to the poor influence of B ligand, we don't totally agree, since the behaviour is consistent with the electronic distribution in the LUMO (see supporting information Table S4) where a high % is concentrate in NN ligand, but there is slightly a contribution of CN, therefore, the LUMO energies of the complexes that have B ligand are different.

  1. In some systems, the 3MC state is more stable than the 3MLCT one. If, as describe in the text “optimized structures of 3MC states were also determined, considering a triplet distorted geometry by a gradual elongation of the metal-ligand bond lengths as described in the literature.”, authors should better explain because this inversion of stability is present only in some systems and should correlate with ligands.

The stabilized 3MC over 3MCLT is observed for 3B and 3C, which is mainly attributed to the effect of the cyclometallating ligand. However, the same behaviour is not observed for 3D (complex similar to 3B based on the electronic nature). To find an explanation, we calculate the root mean square deviation (RMSD), which determines the difference between two sets of coordinate values; in this case between the 3MLCT and 3MC excited states (as explained in comment 3 of referee 2) and based on this value we propose a possible reason for the observed behaviour (see sentences highlighted in green in section 3.5).

Minor revision

3MC optimized geometry is crucial to determine the properties of the different complexes. I suggest to explicitly add in the computational details how these geometries are obtained and not limited this explanation to a couple of references.

These details have been included in the new version of the manuscript (section 2, highlighted in cyan).

Round 2

Reviewer 3 Report

The authors modified the manuscript with the inclusion of all suggestions and dissolved my doubts. The quality of the study is definitively improved. Thus, I suggest the publication in the present form.